# Forecasting Human Trajectory from Scene History

**Mancheng Meng**[1,2]    **Ziyan Wu**[2]    **Terrence Chen**[2]    **Xiran Cai**[1]
**Xiang Sean Zhou**[2]    **Fan Yang**[2*]    **Dinggang Shen**[1,2]
[1]ShanghaiTech University        [2]United Imaging Intelligence
{mengmch,caixr,dgshen}@shanghaitech.edu.cn
{ziyan.wu,terrence.chen,sean.zhou,fan.yang03}@uii-ai.com

## Abstract

Predicting the future trajectory of a person remains a challenging problem, due to randomness and subjectivity of human movement. However, the moving patterns of human in a constrained scenario typically conform to a limited number of regularities to a certain extent, because of the scenario restrictions (*e.g.*, floor plan, roads, and obstacles) and person-person or person-object interactivity. Thus, an individual person in this scenario should follow one of the regularities as well. In other words, a person's subsequent trajectory has likely been traveled by others. Based on this hypothesis, we propose to forecast a person's future trajectory by learning from the implicit scene regularities. We call the regularities, inherently derived from the past dynamics of the people and the environment in the scene, *scene history*. We categorize scene history information into two types: historical group trajectory and individual-surroundings interaction. To exploit these two types of information for trajectory prediction, we propose a novel framework Scene History Excavating Network (SHENet), where the scene history is leveraged in a simple yet effective approach. In particular, we design two components: the group trajectory bank module to extract representative group trajectories as the candidate for future path, and the cross-modal interaction module to model the interaction between individual past trajectory and its surroundings for trajectory refinement. In addition, to mitigate the uncertainty in ground-truth trajectory, caused by the aforementioned randomness and subjectivity of human movement, we propose to include smoothness into the training process and evaluation metrics. We conduct extensive evaluations to validate the efficacy of our proposed framework on ETH, UCY, as well as a new, challenging benchmark dataset PAV, demonstrating superior performance compared to state-of-the-art methods. Code is available at: https://github.com/MaKaRuiNah/SHENet

## 1 Introduction

Human trajectory prediction (HTP) aims to predict a target person's future path from a video clip [5, 9, 19, 20]. This is critical for intelligent transportation, as it enables vehicle to perceive the status of pedestrian in advance so that it can avoid potential collision. Surveillance systems with HTP capability can assist security officers to predict the possible escape path of suspects. Although much work has been done in recent years [1, 9, 12, 14, 18, 35], very few are reliable and generalizable enough to be applied in real-world scenarios, primarily due to two challenges of the task: randomness and subjectivity of human movement. However, in the constrained real world scenarios, the challenges are not absolutely intractable. As shown in Figure 1, given previously-captured video in this scene, the target person's future trajectory (red box) becomes more predictable as the moving pattern of human typically complies with several underlying regularities in this scenario, which the target person

---

*Corresponding author.

36th Conference on Neural Information Processing Systems (NeurIPS 2022).

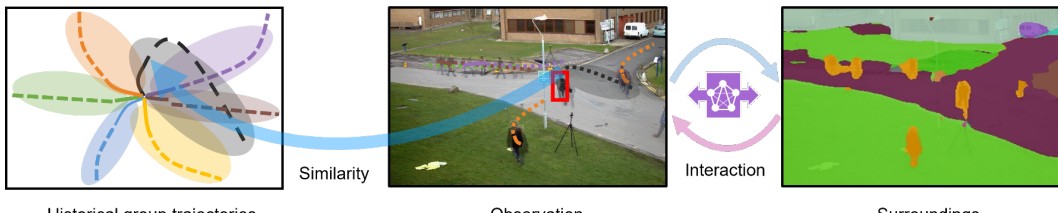

Similarity              Interaction

Historical group trajectories         Observation         Surroundings

Figure 1: Illustration of utilizing scene history: historical group trajectories and individual-surroundings interaction, for human trajectory prediction.

would follow. Therefore, to predict the trajectory, we first need to understand these regularities. We argue that these regularities are implicitly encoded in historical human trajectories (Figure 1 left), individual past trajectory, surroundings, and interaction between them (Figure 1 right), which we refer to as *scene history*.

We separate historical information into two categories: historical group trajectories (HGT) and individual-surroundings interaction (ISI). HGT refers to the group representatives of all historical trajectories in a scene. The reason of resorting to HGT is that, given a new target person in the scenario, his/her path is more likely to share more similarity with one of the group trajectories than any individual instance of the historical trajectories, due to the aforementioned randomness, subjectivity, and regularities. Nevertheless, the group trajectories are less correlated to the individual's past status and the corresponding surroundings, influencing the person's future trajectory as well. ISI is required to more comprehensively leverage history information by extracting contextual information. Few existing methods have taken into account similarity between individual past trajectory and historical trajectories. Most attempts [16, 19, 27, 35] explore only individual-surroundings interaction, where tremendous efforts are spent on modeling the individual trajectory, the semantic information of environment information, and the relationship between them. Although MANTRA [20] models the similarity using the encoders trained in a reconstruction manner and MemoNet [32] simplifies the similarity by storing the intention of historical trajectories, they both perform similarity computation on instance level, instead of group level, making it sensitive to the capability of the trained encoder. Based on the above analysis, we propose a simple yet effective framework, Scene History Excavating Network (SHENet), to leverage HGT and ISI jointly for HTP. In particular, the framework consists of two major components: (i) a group trajectory bank (GTB) module, and (ii) a cross-modal interaction (CMI) module. GTB constructs the representative group trajectories from all historical individual trajectories and provides a candidate path for future trajectory prediction. CMI encodes the observed individual trajectory and the surroundings separately, and models their interaction using a cross-modal transformer for refinement of the searched candidate trajectory.

In addition, to alleviate the uncertainty from two mentioned characteristics (*i.e.*, randomness and subjectivity), we introduce curve smoothing (CS) into the training process and current evaluation metrics, average and final displacement errors (*i.e.*, ADE and FDE), resulting in two new metrics CS-ADE and CS-FDE. Moreover, to facilitate the development of research in HTP, we collect a new challenging dataset with diverse movement patterns, named PAV. The dataset is obtained by choosing the videos with fixed camera view and complex human movement from MOT15 dataset [11].

The contributions of the work can be summarized as follows: 1) We introduce group history to search individual trajectory for HTP. 2) We propose a simple and effective framework, SHENet, to jointly utilize two types of scene history (*i.e.*, historical group trajectories and individual-surroundings interaction) for HTP. 3) We construct a new challenging dataset, PAV; also, one novel loss function and two new metrics, considering randomness and subjectivity in human moving patterns, are proposed to achieve better benchmark HTP performance. 4) We conduct comprehensive experiments on ETH, UCY, and PAV to demonstrate the superior performance of SHENet and the efficacy of each component.

## 2   Related Work

**Unimodal Methods.** Unimodal methods rely on learning the regularity of individual motion from past trajectories in order to predict future trajectories. For example, Social LSTM [1] focuses on

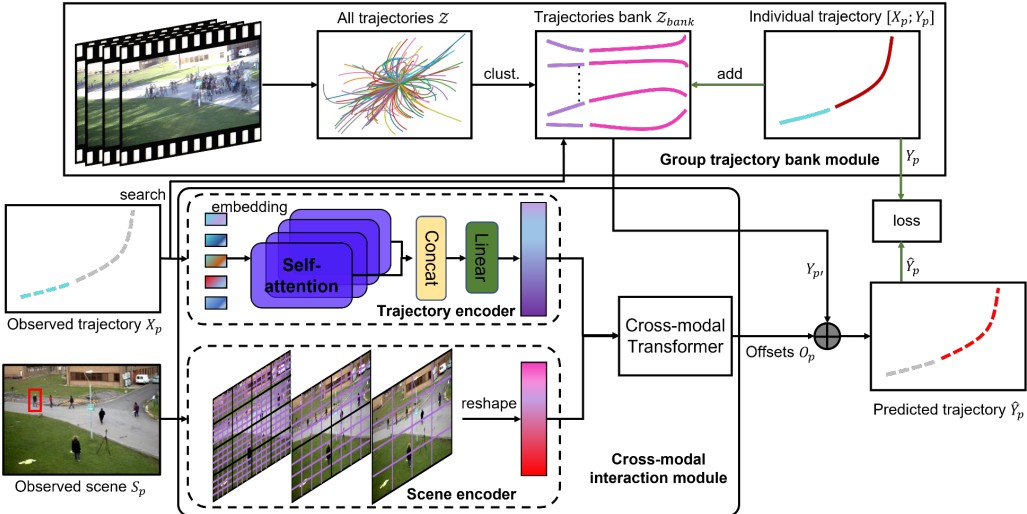

Figure 2: The architecture of SHENet consists of two components: group trajectory bank module (GTB) and cross-modal interaction module (CMI). GTB clusters all historical trajectories into a bank of representative group trajectories and provides candidates for final trajectory prediction. In the training phase, GTB can include the trajectory of a target person into group trajectory bank to extend the expressivity, based on the error of predicted trajectory. CMI takes the past trajectory of a target person and the observed scene as input to a trajectory encoder and a scene encoder for feature extraction, respectively, followed by a cross-modal transformer to effectively model the interaction between the past trajectory and its surroundings and refine the provided candidate trajectory.

modeling the interaction between individual trajectories through a social pooling module. STGAT [8] uses an attention module to learn spatial interactions and assign reasonable importance to neighbors. PIE [25] uses a temporal attention module to calculate the importance of observed trajectories at each time step. Giuliari et al. [6] use the Transformer networks [3, 30] to predict the trajectories of individual people, which weight all observed points in the trajectories according to an attention mechanism. Xu et al. [34] propose a newly designed graph neural network to extract spatial-temporal feature representations of the observed trajectories, along with a knowledge learning module to refine individual-level transferable representations. These methods consider the scene history in two aspects: the individual's past trajectory and the interaction between the individual's past trajectory and its neighboring trajectories.

**Multimodal Methods.** Additionally, multimodal methods examine the influence of environment information on HTP. SS-LSTM [35] proposes a scene interaction module to capture global information of the scene. Trajectron++ [27] uses a graph structure to model the trajectories and interacts with environmental information and other individuals. Zhou et al. [39] propose a context encoder to extract local features for each agent, and a global interaction module to aggregate the local context of individual agents. Liang et al. [16] integrate the interactive modules into a person interaction module (person-person, person-object, and person-scene), and add a person behavior module (person appearance) to encode rich visual information. Different from the RNNs [1, 16, 33, 35] to encode the past trajectories into a hidden state vector, which is prone to loss of information, MANTRA [20] utilizes an external memory to model long-term dependencies. It stores the historical single-agent trajectories in a memory, and encodes the environment information to refine a searched trajectory from this memory.

**Difference from Previous Works.** Both unimodal and multimodal methods use single or partial aspects of the scene history, disregarding the historical group trajectory. In our work, we integrate the information from the scene history in a more comprehensive way and propose dedicated modules to handle different types of information, respectively. The main differences between our method and the previous works, especially memory-based methods [15, 20, 21, 32] and clustering-based methods [28, 29, 36], are as follows: i) MANTRA [20] and MemoNet [32] consider historical individual trajectories, while our proposed SHENet focuses on historical group trajectories, which is more generalizable in different scenarios; In [15], historical trajectories are not stored for search.

---

**Algorithm 1** Group trajectory bank constructing

---

**Input:** Complete trajectory set $\mathcal{Z}$, an input trajectory $Z_p = [X_p; Y_p]$ of person $p$, and our model predicted trajectory $\hat{Y}_p$ of this person

**Output:** Trajectory bank $\mathcal{Z}_{bank}$

1: Initialization : $\mathcal{Z}_{bank} = \emptyset$, added trajectory counter $n \leftarrow 0$, added trajectory threshold $\beta$
2: /* Step 1 : Trajectory bank initialization */
3: Calculate the distance between each pair of trajectories in $\mathcal{Z}$
4: Find medoids for each cluster (via clustering algorithms, K-medoids)
5: Assign each trajectory to the nearest medoid and obtain the initial trajectory bank $\mathcal{Z}_{bank}$ by taking the average of trajectories belonging to the same medoid
6: /* Step 2 : Trajectory search */
7: Initialization : $i \leftarrow 1$, $m \leftarrow Size(\mathcal{Z}_{bank})$
8: **repeat**
9:     Choose the past trajectory $X_i$ of group $i$ from $\mathcal{Z}_{bank}$
10:    Calculate similarity $s_i$ between $X_i$ and $X_p$ based on Equation (1)
11:    $i \leftarrow i + 1$
12: **until** $i > m$
13: Find the most representative trajectory $Y_{p'}$ according to the maximum value of $s$
14: /* Step 3 : Trajectory update */
15: Initialization : $d \leftarrow 0$, distance threshold $\theta$
16: Calculate the distance $d$ between the predicted trajectory $\hat{Y}_p$ with the ground-truth trajectory $Y_p$
17: **if** $d > \theta$ **then**
18:    $n \leftarrow n + 1$
19:    Update the $\mathcal{Z}_{bank}$ by adding the trajectory $Y_p$
20: **end if**
21: **if** $n \geq \beta$ **then**
22:    $n = 0$
23:    Cluster newly added trajectories in $\mathcal{Z}_{bank}$, and merge the clustered trajectories into $\mathcal{Z}_{bank}$
24: **end if**
25: **return** $\mathcal{Z}_{bank}$

---

ii) [28, 36] group person-neighbors for trajectory prediction; [29] clusters the trajectories into fixed number of categories for trajectory classification; Our SHENet generates representative trajectories as reference for individual person trajectory prediction.

# 3   Method

## 3.1   Overview

The architecture of the proposed Scene History Excavating Network (SHENet) is depicted in Figure 2, which consists of two major components: group trajectory bank module (GTB) and cross-modal interaction module (CMI). Formally, given all the trajectories $\mathcal{Z}$ in observed videos of this scene, a scene image $S_p$ and a past trajectory of a target person $p$ in the last $T_{pas}$ time steps, $X_p = \{x_p^t\}_{t=-T_{pas}+1}^{0} \in \mathbb{R}^{T_{pas} \times 2}$, where $x_p^t \in \mathbb{R}^2$ represents the position of $p^{th}$ person at time step $t$, SHENet requires to predict the future positions of the person $\hat{Y}_p = \{\hat{y}_p^t\}_{t=1}^{T_{fut}} \in \mathbb{R}^{T_{fut} \times 2}$ in the next $T_{fut}$ frames, such that $\hat{Y}_p$ is as close to the ground-truth trajectory $Y_p = \{y_p^t\}_{t=1}^{T_{fut}} \in \mathbb{R}^{T_{fut} \times 2}$ as possible. The proposed GTB first condenses $\mathcal{Z}$ as representative group trajectories $\mathcal{Z}_{bank}$. Then, the observed trajectory $X_p$ is used as a key to search a closet representative group trajectory in $\mathcal{Z}_{bank}$ as a candidate future trajectory, $Y_{p'}$. Meanwhile, the past trajectory $X_p$ and a scene image $S_p$ are separately forwarded to the trajectory encoder and the scene encoder to yield the trajectory feature and the scene feature, respectively. The encoded features are fed into the cross-modal transformer to learn the offsets $O_p$ between $Y_{p'}$ and the ground-truth trajectory $Y_p$. By adding $O_p$ to $Y_{p'}$, we obtain our final prediction $\hat{Y}_p$. In the training phase, if the distance of the $Y_p$ to $\hat{Y}_p$ is higher than a threshold, the trajectory of the person (*i.e.*, $X_p$ and $Y_p$) will be added into the trajectory bank $\mathcal{Z}_{bank}$. After training, the bank is fixed for inference.

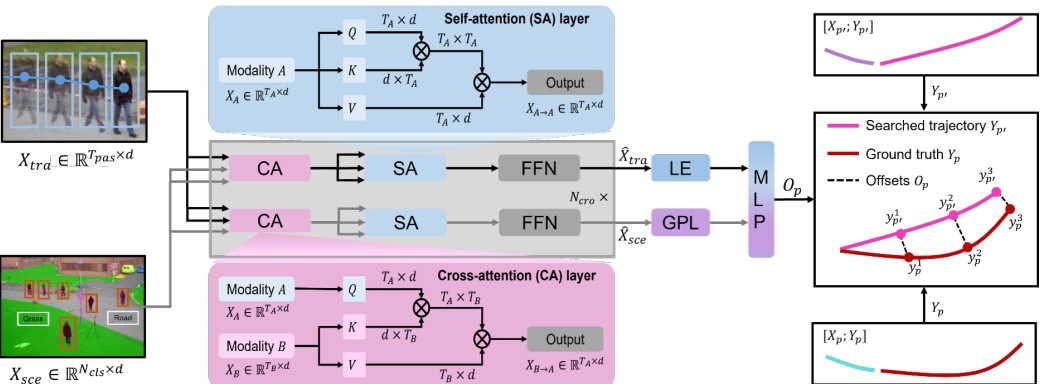

Figure 3: An illustration of cross-modal transformer. The trajectory features and scene features are fed into the cross-modal transformer to learn the offsets between the searched trajectory and the ground-truth trajectory.

## 3.2 Group Trajectory Bank Module

The group trajectory bank module (GTB) is used to construct the representative group trajectories in the scene. The core functions of GTB are bank initialization, trajectory search, and trajectory update.

**Trajectory Bank Initialization.** Due to redundancy of the large amount of recorded trajectories, instead of naively using them, we generate a set of sparse and representative ones as the initial members of the trajectory bank. Specifically, we denote the trajectories in the training data as $\mathcal{Z} = \{Z\}_{i=1}^N$ and split each $Z_i$ into a pair of observed trajectory $X_i$ and future trajectory $Y_i$, so that $\mathcal{Z}$ is separated into an observed set $\mathcal{X} = \{X\}_{i=1}^N$ and a corresponding future set $\mathcal{Y} = \{Y\}_{i=1}^N$. Afterwards, we calculate the Euclidean distance between each pair of trajectories in $\{\mathcal{X}, \mathcal{Y}\}$, and obtain trajectory clusters through the K-medoids clustering algorithm [23]. The initial member of the bank $\mathcal{Z}_{bank}$ is the average of the trajectories belonging to the same cluster (see Algorithm 1, Step 1). Each trajectory in $\mathcal{Z}_{bank}$ represents a moving pattern of a group of people.

**Trajectory Search and Update.** In the group trajectory bank $\mathcal{Z}_{bank}$, each trajectory can be treated as a past-future pair. Numerically, $\mathcal{Z}_{bank} = \{Z_i\}_{i=1}^{|\mathcal{Z}_{bank}|} = \{[X_i; Y_i]\}_{i=1}^{|\mathcal{Z}_{bank}|}$, where $[;]$ represents the combination of a past trajectory and a future trajectory, and $|\mathcal{Z}_{bank}|$ is the number of past-future pairs in $\mathcal{Z}_{bank}$. Given a trajectory $Z_p = [X_p; Y_p]$ of person $p$, we use the observed $X_p$ as a key to compute its similarity score to the past trajectories $\{X_i\}_{i=1}^{|\mathcal{Z}_{bank}|}$ in $\mathcal{Z}_{bank}$ and find a representative trajectory $Y_{p'}, 1 \leq p' \leq |\mathcal{Z}_{bank}|$ according to the maximum similarity score (see Algorithm 1, Step 2). The similarity function can be formulated as:

$$s_i = \frac{X_p \cdot X_i}{\|X_p\|\|X_i\|}, \qquad 1 \leq i \leq |\mathcal{Z}_{bank}| \tag{1}$$

By adding the offsets $O_p$ (see Equation 2) to the representative trajectory $Y_{p'}$, we obtain our predicted trajectory $\hat{Y}_p$ of the observed person (see Figure 2). Although the initial trajectory bank works well for most cases, to improve the generalizability of the bank $\mathcal{Z}_{bank}$ (see Algorithm 1, Step 3), we decide whether to update $\mathcal{Z}_{bank}$ according to the distance threshold $\theta$.

## 3.3 Cross-modal Interaction Module

This module focuses on the interaction between individual past trajectory and the environment information. It consists of two unimodal encoders to learn human motion and scene information, respectively, and a cross-modal transformer to model their interaction.

**Trajectory Encoder.** The trajectory encoder adopts a multi-head attention structure from a transformer network [30] with $N_{tra}$ self-attention (SA) layers. SA layers capture human motion across different time steps with the size of $T_{pas} \times N_{tra} \times d$, and project the motion features from dimension $T_{pas} \times N_{tra}d$ to $T_{pas} \times d$, where $d$ is the embed dimension of the trajectory en-

coder. Thus, we obtain the human motion representation $X_{tra}$ using the trajectory encoder $E_{tra}$: $X_{tra} = E_{tra}(X_p) \in \mathbb{R}^{T_{pas} \times d}$.

**Scene Encoder.** Due to the compelling performance of the pretrained Swin Transformer [17] in feature representation, we adopt it as our scene encoder. It extracts scene semantic features with the size of $N_{cls} \times h \times w$, where $N_{cls}$ ($N_{cls} = 150$ in the pretrained scene encoder) is the number of semantic classes, such as person and road, and $h$ and $w$ are the spatial resolutions. In order to make the following module to conveniently fuse motion representation and environment information, we reshape semantic features from size ($N_{cls} \times h \times w$) to ($N_{cls} \times hw$), and project them from dimension ($N_{cls} \times hw$) to ($N_{cls} \times d$) through a Multi-Layer Perception layer. As a result, we obtain the scene representation $X_{sce}$ using the scene encoder $E_{sce}$: $X_{sce} = E_{sce}(S_p) \in \mathbb{R}^{N_{cls} \times d}$.

**Cross-modal Transformer.** Unimodal encoders extract features from its own modality, disregarding the interaction between human motion and environment information. The cross-modal transformer with $N_{cro}$ layers aims to refine the candidate trajectory $Y_{p'}$ (see Section 3.2) by learning this interaction. We adopt a two-stream structure: one for capturing the important human motion constrained by environment information, and the other for picking out the environment information related to human motion. The cross-attention (CA) layers and self-attention (SA) layers are the main components of the cross-modal transformer (see Figure 3). To capture the important human motion affected by environment and acquire the environment information related to the motion, CA layers treat one modality as query, and the other as key and value to interact with the two modalities. SA layers serve to facilitate better internal connections, which calculate the similarity between an element (query) and other elements (key) in scene-constrained motion or motion-relevant environment information. Thus, we obtain the inter-modal representation ($\hat{X}_{tra}, \hat{X}_{sce}$) via the cross-modal transformer $E_{cro}$: $\hat{X}_{tra} = E_{cro}(X_{tra}, X_{sce}), \hat{X}_{sce} = E_{cro}(X_{sce}, X_{tra})$. To predict the offsets $O_p$ between the searched trajectory $Y_{p'}$ and the ground-truth trajectory $Y_p$, we take the last element ($LE$) $h_{tra} \in \mathbb{R}^d$ of $\hat{X}_{tra}$ and the output $h_{sce} \in \mathbb{R}^d$ of $\hat{X}_{sce}$ after the global pooling layer ($GPL$). Offsets $O_p \in \mathbb{R}^{T_{fut} \times 2}$ can be formulated as follows:

$$O_p = MLP([LE(\hat{X}_{tra}); GPL(\hat{X}_{sce})]) \tag{2}$$

where $[;]$ denotes the vector concatenation, and $MLP$ is the Multi-Layer Perception layer.

### 3.4 Training

We train the overall framework of SHENet end-to-end to minimize an objective function. During the training, since the scene encoder has been pretrained on ADE20K [38], we freeze its segmentation part and update parameters of MLP head (see Section 3.3). Following the existing works [1, 19, 20, 35], we calculate the mean squared error (MSE) between the predicted trajectory and the ground-truth trajectory on ETH/UCY datasets: $\mathcal{L}_{tra} = \frac{1}{T_{fut}} \sum_{t=1}^{T_{fut}} \|y_p^t - \hat{y}_p^t\|_2^2$.

In a more challenging PAV dataset, we use the curve smoothing (CS) regression loss, which serves to reduce the impact of individual bias. It calculates the MSE after trajectory smoothing. CS loss can be formulated as follows:

$$\mathcal{L}_{cs} = \frac{1}{T_{fut}} \sum_{t=1}^{T_{fut}} \|\overline{y}_p^t - \hat{y}_p^t\|_2^2, \qquad \overline{Y}_p = CS(Y_p) = \{\overline{y}_p^t\}_{t=1}^{T_{fut}} \tag{3}$$

where $CS$ represents the function of curve smoothing [2].

## 4 Experiments

### 4.1 Experimental Setup

**Datasets.** We evaluate our method on ETH [24], UCY [13], PAV and Stanford Drone Dataset (SDD) [26] datasets. ETH/UCY datasets are the benchmark commonly used for human trajectory prediction (HTP). The benchmark contains videos from 5 scenes, including ETH, HOTEL, UNIV, ZARA1, and ZARA2 captured at 2.5HZ. Unimodal methods [1, 10, 31, 35] only focus on the trajectory data,

however, multimodal methods [16, 20] need to consider scene information. Since a video from UNIV is not available, [16, 20] use fewer frames than the unimodal methods. We follow [16] to process the data and predict future $n_{fut} = 12$ frames given the observed $n_{pas} = 8$ frames.

Compared to the ETH/UCY datasets, PAV is more challenging with diverse movement patterns, including PETS09-S2L1 (PETS) [4, 11], ADL-Rundle-6 (ADL), and Venice-2 (VENICE), which are captured from static camera and provide sufficient trajectories for the HTP task. We divide the videos into training (80%) and testing (20%) sets, and PETS/ADL/VENICE contain 2,370/2,935/4,200 training sequences and 664/306/650 testing sequences, respectively. We use $n_{pas} = 10$ observed frames to predict future $n_{fut} = 50$ frames, such that we can compare the long-term prediction results from different methods.

Different from ETH/UCY and PAV datasets, SDD is a large-scale datatset captured on the university campus in bird's eye view. It consists of multiple interacting agents (*e.g.*, pedestrians, bicyclists, and cars) and diverse scenes (*e.g.*, sidewalks and crossroads). Following [19], we use past 8 frames to predict future 12 frames. More results conducted on SDD are in the the supplementary material.

**Evaluation Metrics.** For ETH and UCY datasets, we adopt the standard metrics [1, 7, 16, 20, 35] for HTP: Average Displacement Error (ADE) and Final Displacement Error (FDE). ADE is the average $\mathcal{L}_2$ error between predicted trajectories and ground-truth trajectories over all time steps, and FDE is the $\mathcal{L}_2$ error between the predicted trajectories and ground-truth trajectories at the final time step. The trajectories in PAV are with some jittering phenomena (*e.g.*, abrupt and sharp turns). Thus, a reasonable prediction may produce approximately the same error as an unrealistic prediction by using traditional metrics: ADE and FDE (see Figure 7(a)). In order to focus on the pattern and shape of the trajectory itself and also reduce the impact of randomness and

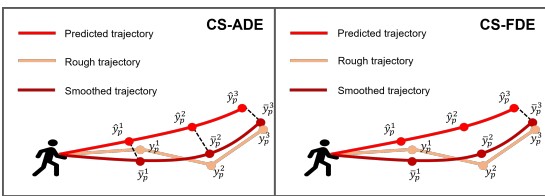

Figure 4: The illustration of our proposed metrics, CS-ADE and CS-FDE.

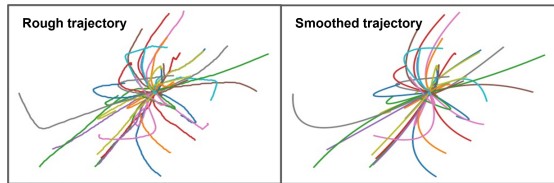

Figure 5: The visualization of some samples after curve smoothing.

subjectivity, we propose the CS-Metric: CS-ADE and CS-FDE (illustrated in Figure 4). CS-ADE is calculated as follows:

$$\text{CS} - \text{ADE} = \frac{\sum_{p=1}^{N} \sum_{t=1}^{T_{fut}} \|\overline{y}_p^t - \hat{y}_p^t\|_2}{N \times T_{fut}}, \qquad \overline{Y}_p = CS(Y_p) = \{\overline{y}_p^t\}_{t=1}^{T_{fut}} \tag{4}$$

where $CS$ is the curve smoothing function defined as the same as our $\mathcal{L}_{cs}$ in Section 3.4. Similar to CS-ADE, CS-FDE calculates the final displacement error after trajectory smoothing:

$$\text{CS} - \text{FDE} = \frac{\sum_{p=1}^{N} \|\overline{y}_p^{T_{fut}} - \hat{y}_p^{T_{fut}}\|_2}{N}, \qquad \overline{Y}_p = CS(Y_p) = \{\overline{y}_p^t\}_{t=1}^{T_{fut}} \tag{5}$$

Figure 5 shows some samples from the training data, converting rough ground-truth trajectories to smooth ones.

**Implementation Details.** In SHENet, the initial size of group trajectory bank is set to $|\mathcal{Z}_{bank}| = 32$. Both the trajectory encoder and the scene encoder have 4 self-attention (SA) layers. The cross-modal transformer is with 6 SA layers and cross-attention (CA) layers. We set all the embed dimensions to 512. For the trajectory encoder, it learns the human motion information with size of $T_{pas} \times 512$ ($T_{pas} = 8$ in ETH/UCY, $T_{pas} = 10$ in PAV). For the scene encoder, it outputs the semantic features with size $150 \times 56 \times 56$. We reshape the features from size $150 \times 56 \times 56$ to $150 \times 3136$, and project them from dimension $150 \times 3136$ to $150 \times 512$. We train the model for 100 epochs on 4 NVIDIA Quadro RTX 6000 GPUs and use the Adam optimizer with a fixed learning rate $1e-5$. More details are in the supplementary material.

## 4.2 Ablation Studies

In Table 1, we evaluate each component of SHENet, including a group trajectory bank (GTB) module, and a cross-modal interaction (CMI) module that contains a trajectory encoder (TE), a scene encoder (SE) and a cross-modal transformer (CMT).

Table 1: Ablation experiments on PAV dataset. ↓ represents that lower is better. Bold/underline denotes the lowest/second low error.

| Method | | | | Evaluation metrics: CS-ADE ↓ / CS-FDE ↓ (in pixels) | | | |
|---|---|---|---|---|---|---|---|
| GTB | TE | SE | CMT | PETS | ADL | VENICE | AVG |
| ✔ | ✗ | ✗ | ✗ | 37.51 / 94.39 | 17.23 / 43.13 | 9.82 / 21.68 | 21.52 / 53.07 |
| ✗ | ✔ | ✗ | ✗ | 38.51 / 104.42 | 19.13 / 51.10 | 11.46 / 24.14 | 23.03 / 59.89 |
| ✗ | ✔ | ✔ | ✗ | 39.53 / 105.41 | 16.13 / 44.49 | 9.77 / 21.65 | 21.81 / 57.18 |
| ✗ | ✔ | ✔ | ✔ | 36.62 / 99.55 | 16.12 / 45.21 | 9.15 / 20.62 | 20.63 / 55.13 |
| ✔ | ✔ | ✔ | ✔ | **34.49 / 78.40** | **14.42 / 38.67** | **7.76 / 18.31** | **18.89/ 45.13** |

**Effect of GTB.** We first investigate the performance of GTB. GTB improves FDE by 21.2% on PETS compared to the CMI (*i.e.*, TE, SE, and CMT), which is a significant improvement, indicating the importance of GTB. However, GTB (1-st row of Table 1) alone is not enough and even performs a little worse than CMI. Thus, we explore the effects of various parts in CMI module.

**Effect of TE and SE.** To evaluate the performance of TE and SE, we concatenate the trajectory features extracted from TE and the scene features from SE together (3-rd row in Table 1), and improve performance on ADL and VENICE with smaller movements, compared to the case of using TE alone. This indicates that the incorporation of environment information into trajectory prediction can improve the accuracy of the results.

**Effect of CMT.** Compared to the third row of Table 1, CMT (4-th row in Table 1) can result in a promising improvement of the model performance. Notably, it performs better than the TE and SE concatenated on PETS, improving by 7.4% over ADE. Compared to GTB alone, the full CMI improves by 12.2% over ADE on average.

## 4.3 Comparison with State-of-the-art Methods

Table 2: Comparison of state-of-the-art (SOTA) methods on ETH/UCY datasets. * represents that we use a smaller set than unimodal methods. The best-of-20 is adopted for evaluation.

| Method | Evaluation metrics: ADE ↓ / FDE ↓ (in meters) | | | | | |
|---|---|---|---|---|---|---|
| | ETH | HOTEL | UNIV* | ZARA1 | ZARA2 | AVG |
| SS-LSTM [35] | 1.01 / 1.94 | 0.60 / 1.34 | 0.71 / 1.52 | 0.41 / 0.89 | 0.31 / 0.68 | 0.61 / 1.27 |
| Social-STGCN [22] | 0.75 / 1.38 | 0.61 / 1.40 | 0.58 / 1.03 | 0.42 / 0.70 | 0.43 / 0.71 | 0.56 / 1.05 |
| MANTRA [20] | 0.70 / 1.76 | 0.28 / 0.68 | 0.51 / 1.26 | 0.25 / 0.67 | 0.20 / 0.54 | 0.39 / 0.98 |
| AgentFormer [37] | 0.52 / 0.84 | 0.15 / 0.22 | 0.34 / 0.72 | **0.18** / 0.33 | 0.16 / 0.30 | 0.27 / 0.48 |
| YNet [19] | 0.47 / 0.72 | **0.12 / 0.18** | 0.27 / 0.47 | 0.20 / 0.34 | **0.15 / 0.24** | 0.24 / 0.39 |
| SHENet (Ours) | **0.41 / 0.61** | 0.13 / 0.20 | **0.25 / 0.43** | 0.21 / **0.32** | **0.15** / 0.26 | **0.23 / 0.36** |

On ETH/UCY datasets, we compare our model with the state-of-the-art methods: SS-LSTM [35], Social-STGCN [22], MANTRA [20], AgentFormer [37], YNet [19]. The results are summarized in Table 2. Our model reduces the average FDE from 0.39 to 0.36, and achieves 7.7% improvement compared with the state-of-the-art method, YNet. Particularly, our model outperforms previous methods significantly on ETH, when there are large movements of trajectories, and improves the ADE and FDE by 12.8% and 15.3%, respectively.

To evaluate the performance of our model in long-term predictions, we conduct experiments on PAV, with $n_{pas} = 10$ observed frames and $n_{fut} = 50$ future frames of each trajectory. Table 3 shows the

performance compared with previous HTP methods: SS-LSTM [35], Social-STGCN [22], Next [16], MANTRA [20], YNet [19]. Compared to the state-of-the-art results of YNet, the proposed SHENet achieves respective 3.3% and 10.5% improvement over CS-ADE and CS-FDE on average. Since

YNet predicts the heatmaps of a trajectory, it performs a little better when there are small movements of trajectories, *e.g.*, VENICE. Nevertheless, our method is still competitive in VENICE, and is much better than others approaches on PETS with large movements and

Table 3: Comparison with SOTA methods on PAV dataset.

| Method | Evaluation metrics: CS-ADE ↓ / CS-FDE ↓ (in pixels) | | | |
|---|---|---|---|---|
| | PETS | ADL | VENICE | AVG |
| SS-LSTM [35] | 39.42 / 107.24 | 16.52 / 50.40 | 10.37 / 23.63 | 22.10 / 60.42 |
| Social-STGCN [22] | 43.40 / 117.85 | 24.34 / 57.22 | 14.42 / 38.66 | 27.39 / 71.24 |
| Next [16] | 37.54 / 98.56 | 16.82 / 46.39 | 8.37 / 19.32 | 20.91 / 54.76 |
| MANTRA [20] | 39.05 / 106.89 | 17.26 / 50.64 | 12.50 / 29.08 | 22.94 / 62.20 |
| YNet [19] | 36.46 / 93.53 | 15.07 / 41.64 | **7.10 / 16.11** | 19.54 / 50.43 |
| SHENet (Ours) | **34.49 / 78.40** | **14.42 / 38.67** | 7.76 / 18.31 | **18.89/ 45.13** |

intersections. In particular, our method improves the CS-FDE by 16.2% on PETS, compared to YNet. We also achieve great improvements using the traditional ADE/FDE metrics. The complete results are shown in the supplementary material.

## 4.4 Analysis

**Distance Threshold $\theta$.** $\theta$ is used to determine the update of trajectory bank. The typical value of $\theta$ is

set according to the trajectory length. When the ground-truth trajectory is longer in terms of pixel, the absolute value of prediction error is usually larger. However, their relative errors are comparable. Thus, the $\theta$ is set to be 75% of the training error when the error converges. In our experiments, we set $\theta = 25$ in PETS, and $\theta = 6$ in ADL. The "75% of the

Table 4: Comparison between different parameters $\theta$ on PAV dataset. Results are the average of the three scenarios.

| $\theta$ | 25% | 50% | 75% | 100% |
|---|---|---|---|---|
| ADE | 22.82 | 20.48 | 18.89 | 20.78 |
| FDE | 58.28 | 51.67 | 45.13 | 54.52 |

training error" is obtained from the experimental result, as shown in Table 4.

**Cluster Number in K-medoids.** We study the effect of setting different number of initial clusters $K$, as shown in Table 5 . We can note that the initial number of clusters is not sensitive to prediction results, especially when the initial number of clusters is 24-36. Therefore, we can set $K$ to 32 in our experiments.

Table 5: Comparison between the initial number of clusters $K$ on PAV dataset.

| K | 24 | 28 | 32 | 36 |
|---|---|---|---|---|
| ADE | 19.28 | 19.23 | 18.89 | 19.17 |
| FDE | 45.48 | 45.59 | 45.13 | 45.62 |

**Analysis for Bank Complexity.** The time complexity of the searching and updating are $O(N)$ and $O(1)$. Their space complexity is $O(N)$. The group trajectory number $N \leq 1000$. The time complexity of the clustering process is $O(\beta^2 + MK\beta)$, and the space complexity is $O(\beta^2 + M\beta)$. $\beta$ is the number of trajectories for clustering. $K$ is the number of clusters, and $M$ is the number of iterations for the clustering methods.

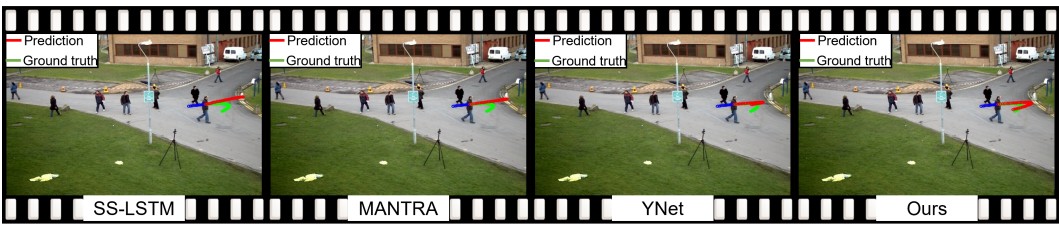

Figure 6: Qualitative visualization of our method and state-of-the-art methods. The blue line is the observed trajectory. The red and green lines show the predicted and ground-truth trajectories.

### 4.5 Qualitative Results.

Figure 6 presents the qualitative results of SHENet and other methods. By comparison, we surprisingly notice that in an extremely challenging situation where a person walks to the curb and turns back (green curve), all other methods do not deal with it well, while our proposed SHENet can still handle it. This should be attributed to the effect of our dedicatedly designed historical group trajectory bank module. In addition, compared to the memory-based method MANTRA [20], we search trajectories of groups, instead of just individuals. This is more versatile and can be applied in more challenging scenes. Figure 7 includes the qualitative results of YNet and our SHENet without/with curve smoothing (CS). The first row presents the results of using MSE loss $\mathcal{L}_{tra}$. Influenced by the past trajectories with some noise (*e.g.*, abrupt and sharp turns), the predicted trajectory points of YNet are gathered together, and cannot present a clear direction, while our method can provide a potential path based on the historical group trajectories. The two predictions are visually distinct, however, the numerical errors (ADE/FDE) are approximately the same. In contrast, the qualitative results of our proposed CS loss $\mathcal{L}_{cs}$ are shown in the second row of Figure 7. We can see that our proposed CS significantly reduces the impact

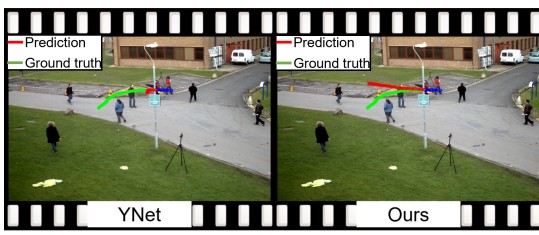

(a) Results of using $\mathcal{L}_{tra}$.

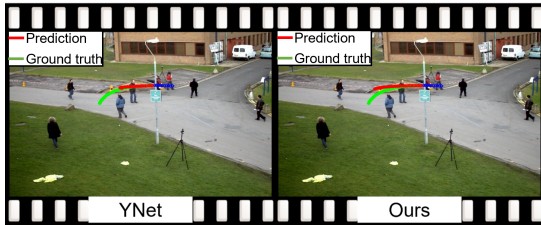

(b) Results of using $\mathcal{L}_{cs}$.

Figure 7: Qualitative visualization without/with CS.

of randomness and subjectivity, and produces reasonable predictions by YNet and our method. More visualization results are provided in supplementary material.

## 5 Conclusion

The paper presents SHENet, a novel method that fully utilizes scene history for HTP. SHENet includes a GTB module to construct a group trajectory bank from all historical trajectories and retrieve a representative trajectory from this bank for an observed person, and also a CMI module (which interacts between human motion and environment information) to refine this representative trajectory. We achieve the SOTA performance on the HTP benchmarks and our method demonstrates significant improvements and generalizaibility in the challenging scenes. However, there are still several unexplored aspects in the current framework, e.g., the process of bank construction currently focusing on only human motion. Future work includes further exploring the trajectory bank using the interact information (human motion and scene information).

## Acknowledgments and Disclosure of Funding

This work is supported by the National Key R&D Program of China (2021ZD0111100) and sponsored by Shanghai Rising-Star Program.

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
