**Supplementary Material**

Section A provides additional details for the method. The experiment details are described in Section B, including experimental setup and results. We also present more qualitative results in Section C.

# A    Method details

## A.1    Cross-modal Interaction

**Trajectory Encoder.** The trajectory encoder aims to learn the human motion representation via a transformer structure. The observed trajectory of person $p$ is $X_p$ with the size of $T_{pas} \times 2$ (*e.g.*, $T_{pas} = 10$ in PAV). We obtain the human motion with the size of $10 \times 4 \times 512$ using $N_{tra} = 4$ self-attention layers, where 512 is the embed dimension. We reshape the motion features from size $10 \times 4 \times 512$ to $10 \times 2048$ and project it from dimension $10 \times 2048$ to $10 \times 512$. Thus, we obtain the human motion representation $X_{tra}$ on PAV dataset with the size of $10 \times 512$.

**Scene Encoder.** The scene encoder is to extract the environment information. The scene semantic features are with the size of $150 \times 56 \times 56$, where 150 is the number of semantic classes. We reshape the semantic features from size $(150 \times 56 \times 56)$ to $(150 \times 3, 136)$, and project them from dimension $(150 \times 3, 136)$ to $(150 \times 512)$ through a Multi-Layer Perception layer. As a result, we obtain the scene representation $X_{sce}$ with the size of $150 \times 512$.

**Cross-modal Transformer.** The cross-modal transformer with $N_{cro} = 6$ layers aims to learn the interaction between human motion and environment information. We obtain the inter-modal representation $(\hat{X}_{tra}, \hat{X}_{sce})$ via the cross-modal transformer. The size of $\hat{X}_{tra}$ and $\hat{X}_{sce}$ are $10 \times 512$ and $150 \times 512$, respectively. We take the last element $(LE)$ $h_{tra} \in \mathbb{R}^{512}$ of $\hat{X}_{tra}$ and the output $h_{sce} \in \mathbb{R}^{512}$ of $\hat{X}_{sce}$ after the global pooling layer $(GPL)$, and get the offsets $O_p \in \mathbb{R}^{50 \times 2}$ $(T_{fut} = 50$ in PAV) through a Multi-Layer Perception layer.

## A.2    Training

We use the curve smoothing (CS) regression loss $\mathcal{L}_{cs}$ to reduce the impact of randomness and subjectivity. The quadratic Bézier curve is adopted to smooth the trajectory, which can be formulated as follows:

$$CS(Z_p, t) = (1 - t)^2 z_p^{-T_{pas}+1} + 2(1 - t)t z_p^{T'} + t^2 z_p^{T_{fut}}, \qquad 0 \le t \le 1 \tag{1}$$

where $z_p^{-T_{pas}+1}$ is the starting point of the trajectory, and $z_p^{T_{fut}}$ is the destination of the trajectory. $z_p^{T'}$ is the control point of this trajectory, which can be calculated as follows:

$$z_p^{T'} = (1 - t)z_p^{-T_{pas}+1} + t z_p^{T_{fut}} \tag{2}$$

Then we divide the time period $t \in [0, 1]$ equidistantly, and get the smoothed trajectory $\overline{Z}_p = [\overline{X}_p; \overline{Y}_p] = \{\overline{z}_p^t\}_{t=-T_{pas}+1}^{T_{fut}}$.

# B    Experiments

## B.1    Experimental Setup

**Dataset Details.** ETH/UCY datasets are the benchmark commonly used for human trajectory prediction. The benchmark contains videos from 5 scenes, including ETH, HOTEL, UNIV, ZARA1, and ZARA2. Following [19], we sample the frames at 2.5 HZ and predict future $n_{fut} = 12$ frames given the observed $n_{pas} = 8$ frames. We use the preprocessed data provided by YNet [19], which converts the raw data from word coordinate into image pixel space. We use the leave-one-scene-out strategy with 4 scenes for training and the remaining scene for testing. PAV is a more challenging dataset with diverse movement patterns, which includes 3 videos PETS, ADL, and VENICE. We divide the videos into training (80%) and testing (20%) sets, and PETS/ADL/VENICE contain

2,370/2,935/4,200 training sequences and 664/306/650 testing sequences, respectively. We use observed $n_{pas} = 10$ frames to predict future $n_{fut} = 50$ frames.

**Evaluation Metrics.** For ETH and UCY datasets, we adopt the standard metrics (*i.e.*, ADE and FDE). Due to the limitations discussed in Section 4.1, we introduce curve smoothing (CS) into current metrics on PAV dataset, and thus we propose CS-ADE and CS-FDE. The curve smoothing function is defined as the same as Equation 1.

## B.2 Experimental Results

**PAV without CS.** We conduct experiments on PAV using the traditional ADE/FDE metrics. Table 1 shows the quantitative result of our method and previous human trajectory prediction methods. Compared to the state-of-the-art results of YNet, the proposed SHENet achieves 5.1%/4.0% improvement over ADE/FDE on average. In particular, our method improves the FDE by 13.6% on PETS.

Table 1: Comparison with SOTA methods on PAV dataset.

| Method | Evaluation metrics: ADE ↓ / FDE ↓ (in pixels) | | | |
| --- | --- | --- | --- | --- |
| | PETS | ADL | VENICE | AVG |
| SS-LSTM [35] | 57.75 / 120.23 | 24.84 / 57.03 | 116.77/ 36.37 | 33.12 / 71.21 |
| Social-STGCN [22] | 63.76 / 159.30 | 31.29 / 73.09 | 19.38 / 43.13 | 38.14 / 91.84 |
| Next [16] | 51.78 / 109.58 | 24.14 / 60.06 | 12.38 / **25.96** | 29.43 / 65.20 |
| MANTRA [20] | 49.87 / 110.14 | 25.78 / 58.12 | 16.79 / 39.50 | 30.81 / 69.25 |
| YNet [19] | 53.46 / 117.81 | **21.95 / 45.88** | **12.29** / 26.61 | 29.23 / 63.43 |
| SHENet (Ours) | **46.31 / 101.74** | 22.46 / 50.71 | 14.43 / 30.21 | **27.73 / 60.89** |

**ETH/UCY without Video Data.** We also conduct the experiments on ETH/UCY without using the video data, shown in Table 2. Since MANTRA didn't conduct experiments on ETH/UCY, we use the results of MANTRA reported in the work [31].

Table 2: Comparison of state-of-the-art (SOTA) methods on ETH/UCY datasets. The best-of-20 is adopted for evaluation.

| Method | Evaluation metrics: ADE ↓ / FDE ↓ (in meters) | | | | | |
| --- | --- | --- | --- | --- | --- | --- |
| | ETH | HOTEL | UNIV | ZARA1 | ZARA2 | AVG |
| Social-STGCNN [22] | 0.64 / 1.11 | 0.49 / 0.85 | 0.44 / 0.79 | 0.34 / 0.53 | 0.30 / 0.48 | 0.44 / 0.75 |
| MANTRA [20] | 0.48 / 0.88 | 0.17 / 0.33 | 0.37 / 0.81 | 0.27 / 0.58 | 0.30 / 0.67 | 0.32 / 0.65 |
| YNet [19] | 0.28 / 0.33 | 0.10 / 0.14 | 0.24 / 0.41 | 0.17 / 0.27 | 0.13 / 0.22 | 0.18 / 0.27 |
| MemoNet [32] | 0.40 / 0.61 | 0.11 / 0.17 | 0.24 / 0.43 | 0.18 / 0.32 | 0.14 / 0.24 | 0.21 / 0.35 |
| SHENet (Ours) | 0.37 / 0.58 | 0.17 / 0.28 | 0.26 / 0.43 | 0.21 / 0.34 | 0.18 / 0.30 | 0.24 / 0.39 |

From Table 2, we can note that our method achieves the comparable performance without using the video data.

**SDD.** We also report the performance of different methods on SDD in Table 3. It shows that our model performs a little bit worse than the results of YNet [19] and MemoNet [32]. Nevertheless, our

Table 3: Comparison of state-of-the-art (SOTA) methods on SDD dataset. The best-of-20 is adopted for evaluation.

| Method | MANTRA [20] | PECNet [18] | YNet [19] | MemoNet [32] | SHENet (Ours) |
| --- | --- | --- | --- | --- | --- |
| ADE | 8.96 | 9.96 | 7.85 | 8.56 | 9.01 |
| FDE | 17.76 | 15.88 | 11.85 | 12.66 | 13.24 |

method performs better than previous baselines (such as PECNet [18], MANTRA [20]). Consequently, our method can achieve reasonable performance in bird-eye-view scenario.

## B.3 Analysis

**Distance Threshold** $\theta$. $\theta$ is used to determine the update of trajectory bank. The typical value of $\theta$ is

set according to the trajectory length. When the ground truth trajectory is longer in terms of pixel, the absolute value of prediction error is usually larger. However, their relative errors are comparable. Thus, the $\theta$ is set to be 75% of the training error when the error converges. In our experiments, we set $\theta = 25$ in PETS, and $\theta = 6$ in ADL. The "75% of the training error" is obtained from the experimental result, shown in Table 4.

Table 4: Comparison between different parameter $\theta$ on PAV dataset. Results are the average of the three scenarios.

| $\theta$ | 25% | 50% | 75% | 100% |
|---|---|---|---|---|
| ADE | 22.82 | 20.48 | 18.89 | 20.78 |
| FDE | 58.28 | 51.67 | 45.13 | 54.52 |

**Different K in K-medoids.** We study the effect of setting different number of initial clusters $K$, shown in Table 5 . We can note that the initial number of clusters is not sensitive to the prediction results, especially when the initial number of clusters is 24-36. Therefore, we can set $K$ to 32 in our experiments.

Table 5: Comparison between the initial number of clusters $K$ on PAV dataset.

| K | 24 | 28 | 32 | 36 |
|---|---|---|---|---|
| ADE | 19.28 | 19.23 | 18.89 | 19.17 |
| FDE | 45.48 | 45.59 | 45.13 | 45.62 |

**Analysis for Bank Complexity.** The time complexity of the searching and updating are $O(N)$ and $O(1)$. Their space complexity is $O(N)$. The group trajectory number $N \leq 1000$. The time complexity of the clustering process is $O(\beta^2 + MK\beta)$, and the space complexity is $O(\beta^2 + M\beta)$. $\beta$ is the number of trajectories for clustering. $K$ is the number of clusters, and $M$ is the number of iterations for clustering methods.

# C   Qualitative Results

Figure 1 presents the visualization of clustered trajectories. We can see that similar trajectories are clustered into the same group. Each group trajectory represents a moving pattern of a group of people. Figure 2 illustrates the qualitative results of SHENet and other methods. Our method is superior to others when it comes to human crossing intersections (*e.g.*, the first row in Figure 2) or turning (*e.g.*, the second row in Figure 2). Figure 3 includes the qualitative results of our SHENet without/with curve smoothing (CS). The first row presents the results of using MSE loss $\mathcal{L}_{tra}$. We can note that our SHENet can not provide a correct path from scene history, since there exists some noise (*e.g.*, abrupt and sharp turns) in past trajectories. Thus to reduce the impact of noise, we use the CS loss $\mathcal{L}_{cs}$ to train our model. In contrast, the qualitative results of using $\mathcal{L}_{cs}$ are shown in the second row of Figure 3. We can see that the proposed CS significantly reduces the impact of randomness and subjectivity, and produces reasonable predictions by our method.

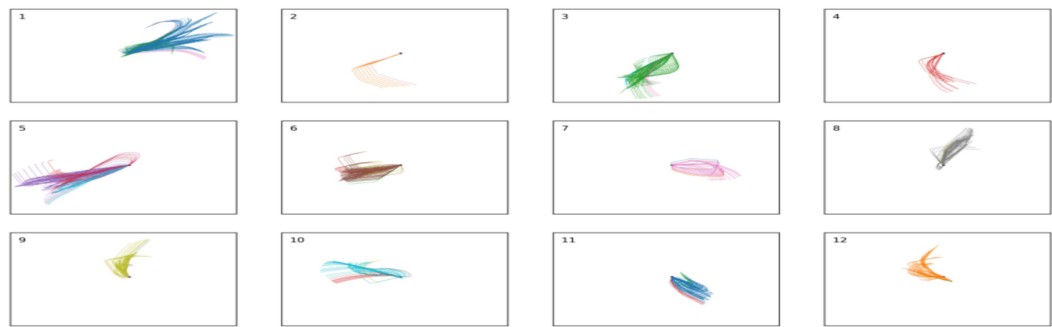

Figure 1: Visualization of clustered trajectories (12 clusters). Each group trajectory is the average of the trajectories belonging to the same cluster.

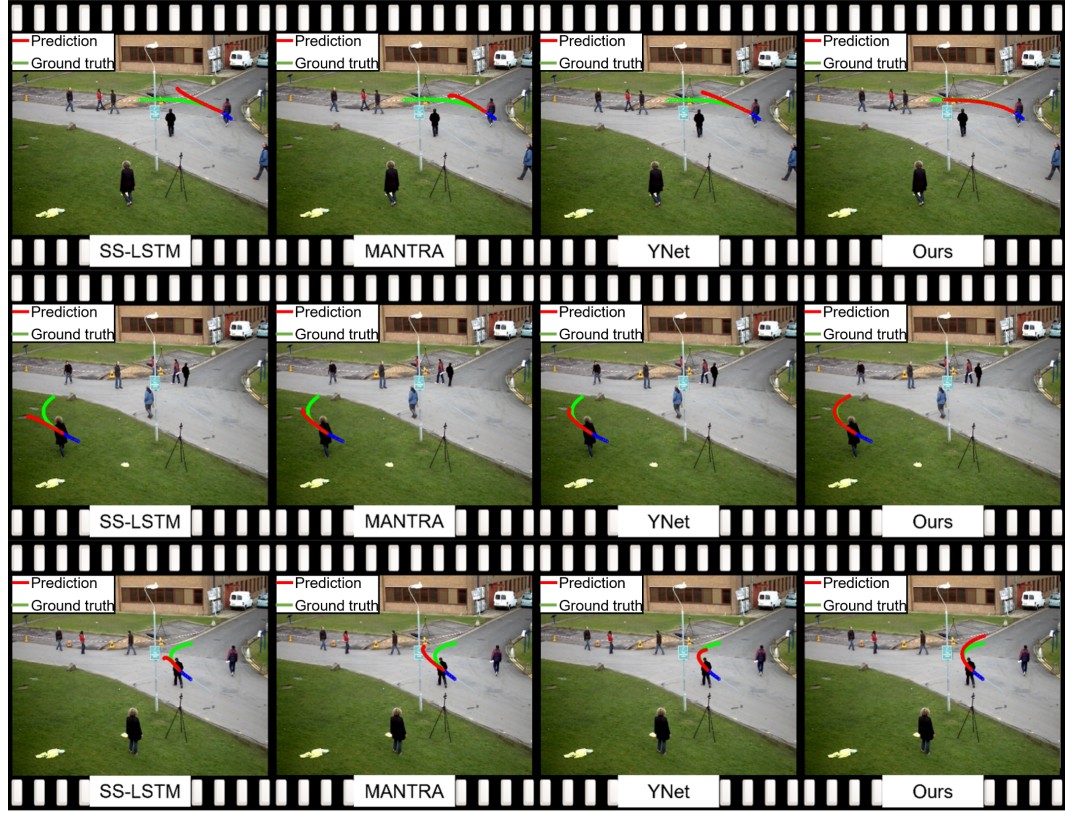

Figure 2: Qualitative visualization of our method and state-of-the-art methods. The blue line is the observed trajectory. The red and green lines show the predicted and ground truth trajectories.

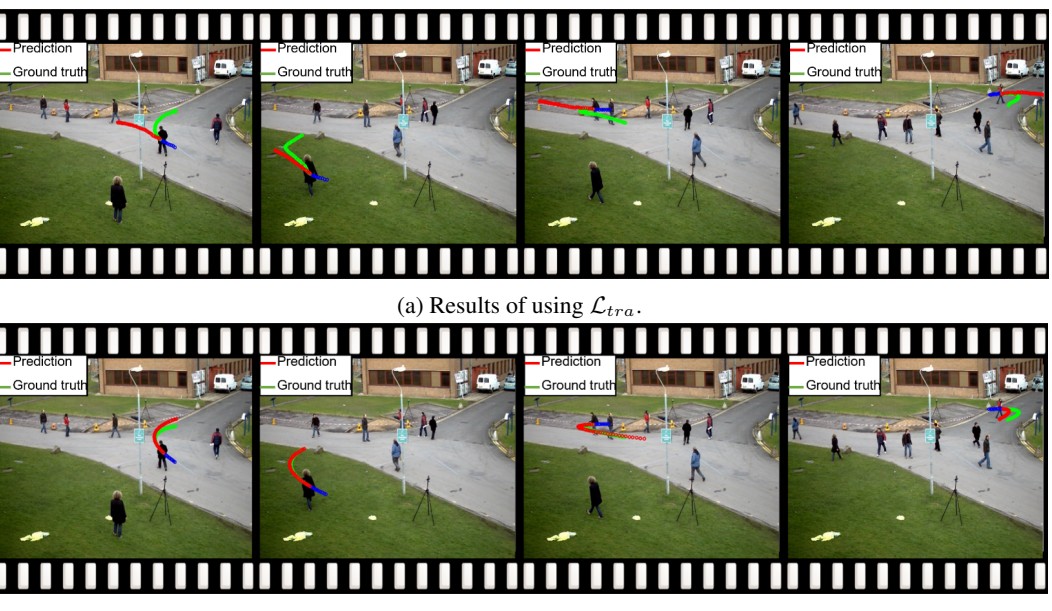

(a) Results of using $\mathcal{L}_{tra}$.

(b) Results of using $\mathcal{L}_{cs}$.

Figure 3: Qualitative visualization of our SHENet without/with CS. The results of using $\mathcal{L}_{tra}$ give a wrong destination, while the results of using $\mathcal{L}_{cs}$ produce reasonable predictions.