# OpenReview forum: "Forecasting Human Trajectory from Scene History"
_NeurIPS.cc/2022/Conference — NeurIPS 2022 Accept_

### Official Review · Reviewer_ZK4Z · 2022-07-09

**Rating:** 6
**Confidence:** 5
**Soundness:** 2 fair
**Presentation:** 3 good
**Contribution:** 2 fair

**Summary:**

In this paper, a method named Scene History Excavating Network (SHENet) is proposed for pedestrian trajectory prediction. The history information is divided into two categories: historical group trajectories and individual-surroundings interaction. Specifically, a group bank module and an interaction module are designed to explore these two information. In addition, the curve smoothing strategy is utilized to alleviate the individual bias. The proposed method can achieve SOTA performance on ETH/UCY as well as a new PAV dataset.

**Questions:**

Concerns:
1. Memory Network is well used in many relevant tasks, i.e., trajectory prediction [1-3], question-answering, anomaly detection, etc, a subsection of related work should be included. Only one paper [4] is mentioned to tell the differences. The approach of trajectory clustering (group idea) is also not new [5-8] here. The similarity function (Eq. (1)) is the same as [4], it should be noted that it is adopted from other papers. In addition, the Transformer structures, including cross-attention and self-attention are also widely used in the trajectory prediction task. Thus, in my opinion, the novelty is not significant.

2. Some key details are missing.
(a). Based on my knowledge, the original videos of ETH/UCY are not aligned with the pedestrian coordinates, so how to align the scene with the trajectory? Or is it directly operated in the feature-level for feature fusion?
(b). In Line 4 of Algorithm 1, the K is initialized as 32 (Line 211), what's the reason/insight of setting this variable equals to 32. It's a really important hyper-parameter since it represents the initial cluster number. I couldn't see any analysis or experiments about this variable.
(c). In Line 155 of Sec. 3.3, the Swin Transformer is applied for extracting scene semantic features, but the number of semantic classes is 150. I assume it is pre-trained here, but why use150 in the trajectory prediction task? I don't think it is reasonable to directly use150 semantic classes here. Details and explanations are missing here.
(d). As for the results of ETH/UCY, some of these baselines are stochastic, the results are usually the best of 20. I couldn't see any stochastic properties in this proposed method, so are the results deterministic or stochastic?

3. Other questions.
(a). The bank is fixed while inference, how to deal with the situation when there is a novel/unseen trajectory (pattern) when testing?
(b). In Eq. (1), the Cosine Similarity is used there, the range of coordinates is not mentioned, usually we normalized the coordinates of ETH/UCY into $[-1, 1]$, which makes the range of $s_{i}$ is $[-1, 1]$. When $s_{i}=-1$, these two trajectories are totally opposite, how to define this kind of similarity? Is it similar or different? I think more explanations should be included here.
(c). What's the meaning/insights of introducing curve smoothing (CS)? If using CS loss could have better performance, does it mean the trajectories in this dataset are more "stable"? Will using CS loss affect the generalization ability of the model itself, only making smooth predictions? Is that prior knowledge?

4. Insufficient experiments.
(a).  No experiment to support the necessity of using CS loss.
(b).  When memory mechanism is used, I think the analysis of complexity/cost is necessary.
(c). The study of Size($Z_{bank}$) is missing, how to initialize the memory bank, the changing process of this memory bank is necessary to support the whole method.


Minors:
1. Line 2, Line 6 and Line 14 of Algorithm 1, $/*$ may need to adjust.
2. Line 7 of Algorithm 1, I believe Size($Z_{bank}$) is K.
3. In References, the names of conferences or journals, i.e. [1] IEEE or IEEE/CVF, are not consistent.

[1] Remember Intentions: Retrospective-Memory-based Trajectory Prediction. CVPR 2022
[2] Graph-Based Spatial Transformer With Memory Replay for Multi-Future Pedestrian Trajectory Prediction. CVPR 2022
[3] Multiple Trajectory Prediction of Moving Agents with Memory Augmented Networks. IEEE TPAMI 2020
[4] Mantra: Memory Augmented Networks for Multiple Trajectory Prediction. CVPR 2020
[5] Recursive Social Behavior Graph for Trajectory Prediction. CVPR 2020
[6] Group lstm: Group Trajectory Prediction in Crowded Scenarios. ECCV Workshop 2018
[7] Three Steps to Multimodal Trajectory Prediction: Modality Clustering,Classification and Synthesis. ICCV2021
[8] PoPPL: Pedestrian Trajectory Prediction by LSTM With Automatic Route Class Clustering. IEEE TNNLS 2020

**Limitations:**

No limitations described in the main body paper as well as the supplementary material.

**Strengths And Weaknesses:**

Strengths:
- Clear-written and well-organized.
- Promising performance.
- A new complex trajectory dataset.


Weaknesses: (See Questions for detailed comments)
- Lack of novelty.
- Missing technique details.
- Insufficient experiments.

---

> ### Author Response · Authors · 2022-08-02
> **Response to Reviewer ZK4Z**
>
> Thank you very much for your detailed review.
>
> Q1-1: Memory networks. A1-1: Our proposed method is not inspired by or similar in architecture with memory network. The only similarity is the underlying idea of storing historical trajectories. The [1] and [2] are officially published at CVPR, Jun 2022, which is after the submission deadline of NeurIPS2022, thus we did not discuss these two papers. We will add them to related works in revision. [1] is totally different from our method: 1) The overall framework is different.[1] first predicts the destination, then fulfills the trajectory to the destination. In contrast, our method directly estimates the future trajectory and leverages the individual-surroundings interaction to refine the trajectory. 2)The historical trajectories are stored in a different manner. In [1],each individual trajectory is embedded through memory network, however, we explicitly utilize the group trajectory. In [2],the historical trajectories are not stored for search. In addition, the proposed "Memory Replay" aims to improve the temporal consistency of the predicted trajectory.The [3] is an extension of included reference [4]. To ensure the integrality of reference, we will cite it in revision.
>
> Q1-2: Clustering is not new. A1-2: The "group" mentioned in [5-8] is not the same as that in our work in terms of concept and motivation. In [5], a group-based model is designed to explore the interaction among pedestrians, while in our work the “group” means a set of representative trajectories. The work [6] is to predict the group trajectory of people, ours is focused on individual person trajectory prediction utilizing historical group trajectories. [7] clusters the trajectories into fixed number of categories for trajectory classification. But the clustering algorithm in our method is used to generate representative trajectories. [8] uses clustering to reduce the complexity of trajectories to alleviate the difficulty in training LSTM for trajectory classification. In contrast, we cluster trajectories to reduce the trajectory search space.
>
> Q1-3: Similarity function. A1-3: The cosine similarity is commonly used in machine learning to measure distance and it is not invented in [4]. Thus, we do not note it is adopted from [4].
>
> Q1-4: Transformer structures. A1-4: The novelty and contributions of our work lie in the insight motivation and underlying principle of designing network structure rather than the adoption of transformer structure for trajectory prediction.  In addition, we request the reviewer to specify the transformer-based works and point out the similarities. It is not fair to say our work is not novel just because the transformer structure is used.
>
> Q2-1: How to align the scene with the trajectory? A2-1: We use the preprocessed data provided by YNet(Ref.[18]  in paper) . It has converted the data from word coordinate into image pixel space.
>
> Q2-2: The reason/insight for setting this variable to 32. A2-2: We note that the initial number of clusters is not sensitive to the prediction results, especially when the initial number of clusters is 24-36. The results are listed below.
>
> |K|24|28|32|36|
> |--|--|--|--|--|
> |ADE|19.28|19.23|18.89|19.17|
> |FDE |45.48|45.59|45.13|45.62|
>
> Q2-3: Why 150 semantic classes? A2-3: The main reasons are two-fold: 1)The category of each scenario is unknown beforehand; 2)Our cross-modal interaction module can reweight the semantic features for human motion (Line 164-172).
>
> Q2-4: Deterministic or stochastic? A2-4: Explained in Reviewer bcbq A1.
>
> Q3-1: Unseen trajectory? A3-1: Explained in Reviewer tAHR A2.
>
> Q3-2: The coordinates and similarity. A3-2: As mentioned in A2-1, we use the preprocessed data from YNet and the word coordinates are converted into pixels. Thus, we can directly use the cosine similarity in the image space.
>
> Q3-3: Meaning/insights of curve smoothing (CS)? A3-3: Due to the randomness and subjectivity, human trajectory visually appears as an irregular curve. The irregularity makes it hard for the model to learn the past-future trajectory pair. We propose the CS loss to alleviate this in model learning.
>
> Q4-1: Experiments to support CS loss. A4-1: As mentioned in A3-3, we conduct the experiments using different methods with/without CS loss on a complex dataset PAV (Tab.3 in paper, and Tab.1 in SM). The qualitative results are in Fig.7 of the paper and Fig.3 in SM.
>
> Q4-2: Complexity. A4-2: The time complexity of the searching and updating are O(N) and O(1). Their space complexity is O(N). The group trajectory number N <=1000. The time complexity of the clustering process is O($β^2 + MKβ$), and the space complexity is O($β^2+Mβ$). β is the number of trajectories for clustering, K is the number of clusters, and M is the number of iterations for clustering methods.
>
> Q4-3: The bank size, initialization, and update. A4-3: As mentioned in A4-2, the bank size is up to 1000. The initialization and update processes are described in Alg.1.

---

> > ### Comment · Reviewer_ZK4Z · 2022-08-08
> > **Thanks for Response**
> >
> > Thanks for the response, most of my concerns are well-addressed and I am planning to increase my rating.
> >
> > But before I increase the score, I would like to see the revised paper uploaded. I will increase the score once all the things are included in the revised paper.

---

> > > ### Author Response · Authors · 2022-08-09
> > > **Follow up response to Reviewer ZK4Z**
> > >
> > > Thank you very much for the feedback. Due to the page limit, we add the changes in both main text and supplementary materials. The brief summary of these changes is listed here:
> > >
> > > 1. The missing related works are added in the manuscript (Line 96-102).
> > > 2. Add some details about the processed data of ETH/UCY in the supplementary material (Line 36-38).
> > > 3. Add the analysis about the initial bank size $K$ in the supplementary material (Line 63-64).
> > > 4. Add the sampling protocol in the table caption (Tab.2 in our manuscript, Tab.1-3 in supplementary material).
> > > 5. Add the analysis of bank complexity in the supplementary material (Line 64-68).
> > > 6. Add some experiments and analysis according to the reviewers’ suggestions in the supplementary material (Section B.2 and B.3).

---

> > > > ### Comment · Reviewer_ZK4Z · 2022-08-09
> > > > **Rating and Minors**
> > > >
> > > > Thank you for uploading the revised the paper. I've checked the manuscript and increased my rating. I highly encourage authors to include several more lateset works in this year CVPR, since there are some highly relevant papers in the proceedings. i.e. [1][2][3] etc.
> > > >
> > > > Also, there are some typos in the manuscript, please double revise.
> > > >
> > > > [1] Adaptive Trajectory Prediction via Transferable GNN. CVPR 2022
> > > > [2] How Many Observations are Enough? Knowledge Distillation for Trajectory Forecasting. CVPR 2022
> > > > [3] HiVT: Hierarchical Vector Transformer for Multi-Agent Motion Prediction. CVPR 2022

---

### Official Review · Reviewer_bcbq · 2022-07-10

**Rating:** 6
**Confidence:** 4
**Soundness:** 3 good
**Presentation:** 3 good
**Contribution:** 2 fair

**Summary:**

This paper focus on the task of human trajectory prediction. The authors stress the importance of scene history, including historical group trajectories and individual-surroundings interaction, and design a clustering & update procedure to maintain a group trajectory bank, which is used to generate trajectory prediction proposals. A cross modal transformer fuse the feature of observed trajectory and scene information, which is then used to predict the offset from the trajectory proposals. The authors conduct intensive experiments on ETH, UCY, and PAV datasets.

**Questions:**

None

**Limitations:**

About the ethical impact: acquiring human trajectory dataset may violate privacy under some circumstances. The authors may consider to add the potential negative societal impact.

**Strengths And Weaknesses:**

Strengths:
1. The paper is well-organized and easy to follow. The authors state their motivation quite clear.
2. The novelty and originality are good. The group trajectory bank module is innovating.
3. The ablative study is comprehensive and convincing. The results shows that the main module - group trajectory bank, improve the performance by a non-trivial margin.

Weaknesses:
1. In the experiment section of ETH/UCY dataset, the authors didn't states the setting clear: what is the sampling protocol when calculating ADE/FDE? Standard protocol is best-of-20. Also, as the architecture itself is not stochastic, it'll be better if deterministic prediction ADE/FDE is added.
2. The performance of previous work in the table is different from the results from the original paper. For example, Y-net achieve 0.18/0.27 according to the original paper, but in table 2 it becomes 0.24/0.39. Is it due to different experiment setting or different implementation?

---

> ### Author Response · Authors · 2022-08-02
> **Response to Reviewer bcbq**
>
> Thank you very much for your kind comments and feedback. We will explain your concerns point by point.
>
> Q1: What is the sampling protocol when calculating ADE/FDE?
>
> A1: We use the standard protocol (best-of-20). In order to make comparisons as fair as possible, several methods are used and listed in Tab. 2, including both stochastic methods (Social-STGCN, AgentFormer, YNet) and deterministic methods (SS-LSTM, MANTRA, SHENet [Ours]). Although stochastic methods may have a little unstable performance because of the sampling process, our compared methods adopt different sampling strategies to reduce the uncertainty. We explain the predicted results of these methods as follows. a) Social-STGCN is a typical probabilistic method to model a probability distribution. During inference, they sample 20 trajectories from the probability distribution. It is easier to sample trajectories with high probability. b) AgentFormer uses a trajectory sampler to generate 20 plausible latent variables. During inference, it samples latent variables from the trajectory sampler and uses them to produce predictions (this process can also be viewed as searching the most probable 20 trajectories). c) YNet aims to find the most probable 20 trajectories with the argsoftmax function to approximate the goal’s and waypoints’ most likely position. In general, many stochastic methods use probabilistic strategies to find plausible trajectories, while MANTRA and ours generate plausible trajectories based on the similarity between the observed trajectory and elements in memory or group trajectory bank. Specifically, MANTRA searches for the top20 hidden states that are most like the observed trajectory features, and then the top20 hidden states are used to produce predictions. Our SHENet obtains the top20 of group trajectories that are similar to the observed trajectory from bank, as well as an interaction module to refine the searched trajectories. The contrast between stochastic methods using probability and deterministic methods using similarity is worth exploring.
>
> Q2: Why are the results different from the original paper?
>
> A2: Please see Reviewer XkVC A3.

---

### Official Review · Reviewer_tAHR · 2022-07-10

**Rating:** 6
**Confidence:** 4
**Soundness:** 3 good
**Presentation:** 3 good
**Contribution:** 4 excellent

**Summary:**

This paper proposes to use the "scene history" for the human trajectory prediction. The "scene history" consists of two parts: historical group trajectories and individual-surroundings interaction. This paper proposes a SHENet to jointly use these two "scene history" clues.

Extensive experiments are conducted on ETH, UCY, and a new, challenging benchmark dataset PAV.


**Questions:**

See [Weakness1] and [Weakness2].

**Limitations:**

No clear limitations and potential negative societal impact.

**Strengths And Weaknesses:**

[Strength1] The proposed method which uses the "scene history" (especially historical group trajectories) is well-motivated.

[Strength2] The basic idea makes sense. Historical group trajectories and individual-surroundings interaction are helpful to understand human behavior.

[Strength3] This paper provides a new challenging dataset.

[Strength4] The proposed method achieve good performance.

[Weakness1] It is suggested to add the experiments on the SDD dataset, which is also an important benchmark for human trajectory prediction.

[Weakness2] For most real-world applications, we can not obtain the scene in which the model will be used. In the proposed method, the bank is fixed for inference after training. How to guarantee the learned bank with training scene can be used for the new environment.

[Weakness3] The reproduction is difficult since no source code is attached in the supp.

---

> ### Author Response · Authors · 2022-08-02
> **Response to Reviewer tAHR**
>
> We greatly appreciate the positive comments and valuable suggestions. We will explain your concerns point by point.
>
> Q1: It is suggested to add the experiments on the SDD dataset.
>
> A1: As explained in Reviewer XkVC A2-1 and Reviewer XkVC A2-3, the SDD dataset is not suitable for the assumption of our method. However, we will conduct experiments on SDD, and include in the revision.
>
> Q2: How to guarantee the learned bank with training scene can be used for the new environment?
>
> A2: Good question. Currently, our method is designed for human trajectory prediction in a constrained environment. This means we assume that the scene is fixed or similar. If the new environment is totally different with our training scene, the trained bank may degrade dramatically. This also is a common problem with many search-based methods (e.g., MANTRA [1]). However, compared to the individual-based methods, our group-based method is more generalizable (e.g., visualization results in Fig. 6).
>
> [1] Mantra: Memory Augmented Networks for Multiple Trajectory Prediction. CVPR 2020

---

> > ### Comment · Reviewer_tAHR · 2022-08-08
> > **Thanks for the response**
> >
> > Thanks very much for your response! There are still two concerns about the experiments on the large-scale dataset such as SDD.
> >
> > 1) I have looked through the revised paper and did not find the results on the SDD dataset. Can you kindly tell me the location or show it in the response?
> >
> > 2) As shown in XkVC A2-3, the history search requires lots of information including the person's state and appearance, background information, and interaction of person-background/person.  If the model fails when some information is missed? In many applications, we only have a bird-eye-view information. Can the proposed method work well?
> >
> > 3) Can you show the performance on ETH_UCY without using the video data? It will help to explore the boundary of the method.

---

> > > ### Author Response · Authors · 2022-08-08
> > > **Follow up response to Reviewer tAHR**
> > >
> > > Thank you very much for your kind comments and feedback.
> > >
> > > Que1: The results conducted on the SDD dataset.
> > >
> > > Ans1: We experimented on the SDD dataset and gave a preliminary result below. Now, we have added this content in the supplementary material (Tab. 2).
> > >
> > > |Method|MANTRA|PECNet|YNet|MemoNet(CVPR'22)[1]|SHENet|
> > > |--|--|--|--|--|--|
> > > |ADE|8.96 |9.96 |7.85|8.56|9.01|
> > > |FDE |17.76 | 15.88 |11.85|12.66|13.24|
> > >
> > > Que2: Can the proposed method work well?
> > >
> > > Ans2: As the results mentioned in Ans1, our model performs a little bit worse than the results of YNet and MemoNet. Nevertheless, our method performs better than previous baselines (such as PECNet, MANTRA). Consequently, our method can achieve reasonable performance in bird-eye-view scenario.
> > >
> > > Que3: Can you show the performance on ETH/UCY without using the video data?
> > >
> > > Ans3: We conduct the experiments on ETH/UCY according to your suggestions. The results are listed below, where the best-of-20 is adopted for evaluation. Since MANTRA didn’t conduct experiments on ETH/UCY, we use the results of MANTRA reported in the work[1].
> > >
> > > |Method|ETH|HOTEL|UNIV|ZARA1|ZARA2|AVG|
> > > |--|--|--|--|--|--|--|
> > > |Social-STGCNN|0.64/1.11|0.49/0.85 |0.44/0.79|0.34/0.53|0.30/0.48 |0.44/0.75|
> > > |MANTRA|0.48/0.88|0.17/0.33|0.37/0.81|0.27/0.58|0.30/0.67|0.32/0.65|
> > > |YNet|0.28/0.33|0.10/0.14|0.24/0.41|0.17/0.27|0.13/0.22|0.18/0.27|
> > > |MemoNet[1]|0.40/0.61|0.11/0.17|0.24/0.43|0.18/0.32|0.14/0.24|0.21/0.35|
> > > |SHENet|0.37/0.58|0.17/0.28|0.26/0.43|0.21/0.34|0.18/0.30|0.24/0.39|
> > >
> > > From the table, we can note that our method achieves the comparable performance without using the video data.
> > >
> > > [1] Remember Intentions: Retrospective-Memory-based Trajectory Prediction. CVPR 2022

---

### Official Review · Reviewer_XkVC · 2022-07-11

**Rating:** 5
**Confidence:** 5
**Soundness:** 2 fair
**Presentation:** 2 fair
**Contribution:** 2 fair

**Summary:**

The authors tackle the problem of human trajectory forecasting with the help of a novel architecture. The architecture has two components, the group trajectory bank module to extract representative group trajectories as the candidate for future path, and the cross-modal interaction module to model the interaction between individual past trajectory and its surroundings for trajectory refinement, respectively. The authors construct a new challenging dataset, PAV for long-term prediction. The article demonstrates the superiority of the proposed model on ET/UCY and the new PAV dataset over state-of-the-art baseline methods. A new set of metrics, curve smoothing ADE/FDE have also been introduced here.

**Questions:**

As also previously mentioned:
- What are the typical values of the trajectory threshold, beta and distance threshold, theta, used as hyper-parameters in the construction of group trajectory bank? Are the hyper-parameter values specific to a dataset?
- What is the exact contribution of the PAV dataset? What are the criteria for the chosen subset of the MOT15 videos? What is the reason for introducing this dataset instead of experimenting on the existing and used datasets, such as Stanford Drone Dataset (SDD), Intersection Drone Dataset (inDD)?
- The reported performance of some SOTA works on the ETH/UCY datasets is much worse than the results of their original papers. Why are the original results not reported in the comparison?

**Limitations:**

The authors did not discuss any limitations or potential negative social impact of the work.

**Strengths And Weaknesses:**

Strengths:
- The work is of moderate originality, with a fair presentation and writing quality. The proposed ideas of group memory and cross-modality fusion are interesting.
- The proposed models seems to be able to improve the prediction performance concerning the reported state-of-the-art results. Nevertheless, there is some lack of clarification in the experiments and evaluation part, especially about the reported SOTA results and how they were obtained.

Weaknesses:
- It is not clear the exact contribution of the PAV dataset; what is the criteria for the chosen subset of the MOT15 videos? What is the reason for using this dataset instead of existing and used datasets, such as NuScenes, Oxford RobotCar, and Cityscapes?
- It makes sense to enhance the loss function with curve smoothing during training; however, is it fair to use curve smoothing only on the ground truth during evaluation, especially for the state-of-the-art works that did not apply CS-loss? Actually, it is not mentioned if the SOTA works were retrained with CS-loss or not, and maybe it would be more convincing to include an evaluation with the default metrics as well as the new metrics.
- The reported performance of some SOTA works on the ETH/UCY datasets is much worse than the results of their original papers. Why are the original results not reported in the comparison?
- It is not mentioned anywhere in the paper what are the typical values of the trajectory threshold, beta and distance threshold, theta, used as hyper-parameters in the construction of group trajectory bank? It is also not said anywhere if the hyper-parameter values are specific to a dataset?
- A comparative study could be performed with different types of clustering methods, used for constructing the group trajectory bank.

Minor comments:
- One minor mistake in row 108: t should start from -T_{pas+1}, instead of -T_{pas}.

---

> ### Author Response · Authors · 2022-08-02
> **Response to Reviewer XkVC**
>
> Thank you very much for your detailed review and feedback. We will address the major concerns below.
>
> Q1: What are the typical values of θ and β? Are the θ and β specific to a dataset?
>
> A1: Distance threshold θ is used to determine the update of trajectory bank $Z_{bank}$. The typical value of θ is set according to the trajectory length. When the ground truth trajectory is longer in terms of pixel, the absolute value of prediction error is usually larger. However, their relative errors are comparable. Thus, the θ is set to be 75% of the training error when the error converges. In our experiments, we set θ = 25 in PETS, and θ = 6 in ADL. The "75% of the training error" is obtained from the experimental result, which is listed in the following table.
>
> |$\theta$| 25\%|50\%|75\%|100\%|
> |--|--|--|--| --|
> |ADE|22.82|20.48|18.89|20.78|
> |FDE |58.28|51.67|45.13|54.52|
>
> Trajectory threshold β specifies the threshold for the number of newly added trajectories in bank $Z_{bank}$ when updating (Alg. 1, Line 21-23). In our experiment, we set β = 200 for both PAV and ETH/UCY.
>
> Q2-1: What is the exact contribution of the PAV dataset?
>
> A2-1: The contributions of PAV are 1) comparing with ETH/UCY, the captured scenario and trajectory are more complex and realistic; 2) the clear spatial information of the surroundings, including people's state and appearance (missing in SDD/inDD) and background layout, is helpful to explore the individual-surroundings interaction for human trajectory prediction. Therefore, the data can better facilitate the trajectory prediction research.
>
> Q2-2: What are the criteria for the chosen subset of the MOT15 videos?
>
> A2-2: Human trajectory prediction is challenging due to randomness and subjectivity. Based on our analysis and to make the task more practical, we restrict the prediction task to a constrained scenario, that is, the chosen video sequences should be captured with a static camera.
>
> Q2-3: What is the reason for introducing PAV instead of experimenting on SDD and inDD?
>
> A2-3: According to the analysis in our paper, given a person’s past trajectory in a scene, the future path can be inferred by refining the searched historical trajectories of the scene using individual-surroundings information. The individual-surroundings consists of the person state and appearance, background information, and interaction of person-background/person. The person's state and appearance are not available in SDD/inDD because of the bird-eye-view and long-distance capture. In contrast, PAV can provide all these information. Therefore, we do not use SDD/inDD but introduce PAV dataset.
>
> Q3: Why are the original results not reported in the comparison?
>
> A3: As mentioned in A2-3, we need the individual-surroundings, where image information is required. ETH/UCY contains trajectories from five scenes: ETH, HOTEL, UNIV, ZARA1 and ZARA2, but video in UNIV is not available. Thus, we use fewer sequences than the original experiments (mentioned in the paper, Line 196). For fair comparison, we reproduce results of all the methods in this dataset.

---

### Meta-Review · Area_Chair_r49p · 2022-08-23

**Recommendation:** Accept
**Confidence:** Certain

**Metareview:**

Initially, the paper received mixed reviews (3456).  The major concerns raised by the reviews were:

1. What is the contribution of the PAV dataset? (XkVC)
2. There should be experiments on existing datasets, e.g. SDD or inDD. (XkVC, tAHR)
3. Is it fair to use curve smoothing on the GT during evaluation? To be fair, other works should be trained with CS loss too. What is the reason for CS, does it affect generalization of the model? No ablation for CS loss (XkVC, ZK4Z)
4. reported results of some SOTA works on ETH/UCY are worse than the published results. Why not use the original reported results? (XkVC, bcbq)
5. what are the settings of the hyperparameters? Is it dataset specific? (XkVC)
6. ablation study on the clustering methods used for constructing the group trajectory bank. (XkVC)
7. The bank is fixed after training -- how to guarantee the learned bank can be used in a new environment, new scene (tAHR, ZK4Z)
8. show results on ETH/UCY w/o using video data (tAHR)
9. can the proposed method work well from bird-eye view or when assumed information is missing? (tAHR)
10. evaluation metric used on ETH/UCY is not clear. stochastic vs deterministic? (bcbq, ZK4Z)
11. novelty is not significant (ZK4Z)
12. Missing details: scene-trajectory alignment, how to set K, how to set no. of semantic classes, range of 13. coordinates for cosine similarity, (ZK4Z)
14. No ablation on memory vs complexity (ZK4Z)
15. No ablation on the memory bank size and initialization (ZK4Z)

The authors wrote a response to address these concerns. All reviewers were satisfied with the response.  Overall, the reviewers found the paper interesting, although the approach is somewhat incremental and more experiments could be added (e.g., on SDD and inDD, as well as ablation studies).  Nonetheless, the final ratings increased to 5666.  The AC agrees with the reviewers and thus recommends accept. The authors should further revise the paper according to the reviews and discussion.

**Award:**

No

---

### Decision · Program_Chairs · 2022-09-14

Accept